# Mentorship to strengthen health system leadership: A case study of the Walungu rural health zone in the eastern Democratic Republic of Congo

Rosine Bigirinama[1,2,3]*, Ghislain Bisimwa[1,2,4], Samuel Makali[1,5], Aimé Cikomola[1], Janvier Barhobagayana[6], Jean-Corneille Lembebu[1], Christian Chiribagula[1], Pacifique Mwene-Batu[1,7], Abdon Mukalay[3], Denis Porignon[8], Albert Tambwe[3]

1 Ecole Régionale de Santé Publique, Université Catholique de Bukavu, Bukavu, Democratic Republic of Congo, 2 School of Medicine, Université Catholique de Bukavu, Bukavu, Democratic Republic of Congo, 3 Ecole de Santé Publique, University of Lubumbashi, Lubumbashi, Democratic Republic of Congo, 4 Centre de Recherche en Sciences Naturelles, Lwiro, Democratic Republic of Congo, 5 Centre de Recherche (CR3) Politiques et Systèmes de Santé- Santé internationale, École de Santé Publique, Université Libre de Bruxelles, Bruxelles, Belgique, 6 Department of Public Health, College of Health Sciences, Université Officielle de Bukavu, Bukavu, Democratic Republic of Congo, 7 School of Medicine, Université de Kaziba, Bukavu, Democratic Republic of Congo, 8 Département des Sciences de la Santé Publique, School of Medicine, Université de Liège, Liège, Belgium

* rosinebigirinama@gmail.com

**Data Availability Statement:** The quantitative dataset supporting the findings of this study is

## Abstract

From 2015 to 2019, the "RIPSEC" program launched a mentorship program, transforming the Walungu health zone, in eastern crisis-affected Democratic Republic of Congo, into a "Learning and Research Zone" (LRZ). As part of the program, a local university was tasked with strengthening the LRZ manager's leadership capacities, including efforts to trouble-shoot challenges related to the proliferation of informal healthcare facilities (IHFs). IHFs are unregulated healthcare structures operating on the fringes of the law, and claiming to offer cheaper, higher-quality care to the local population. This study evaluates the impact of RIPSEC mentorship on leadership development and the performance in the Walungu LRZ, particularly concerning the utilization of integrated curative health services in competition with IHFs. We used a mixed method approach, combining retrospective analysis of some key health indicators before (2014) and during RIPSEC program (2014 vs. 2015–2019), and in-depth qualitative interviews with members of the LRZ management team. Quantitative data were presented as frequencies and proportions. Simple linear regression (p<0.05) measured the influence of IHFs on service use. The LRZ's functionality and performance were assessed using an internal benchmarking approach, with results presented as trend curves. Deductive analysis of interviews allowed for a deeper exploration of quantitative trends. Despite efforts by the LRZ managers to regulate IHFs, these structures negatively impact the use of curative services by diverting patients away from integrated healthcare options. RIPSEC mentorship notably enhanced manager's leadership skills, leading to more effective management. While the use of curative health services slightly increased during the program, rates remained below 50%, and gains were not sustained post-program. RIPSEC

available as supplementary information, along with the interview guide used for qualitative data collection. However, the qualitative interview transcriptions will not be publicly available to protect the confidentiality of the respondents, as per the guidelines approved by our research ethics board. Interested researchers can request access to the qualitative data by contacting the institutional Data Access Committee at recherche. ersp@ucbukavu.ac.cd. The committee will review requests to ensure they meet relevant ethical and legal guidelines before granting access.

**Funding:** The authors received no specific funding for this work.

**Competing interests:** The authors have declared that no competing interests exist.

mentorship has positively impacted leadership and performance in Walungu. However, financial challenges and the persistent influence of IHFs continue to impede the sustainability of these gains. Comprehensive strategies beyond enhancing managerial leadership solely, are necessary.

## Introduction

In contexts of prolonged multifaceted crises, healthcare systems face colossal challenges that progressively undermine their ability to provide quality health services to the population [1, 2]. In 2021, 4.5 billion people worldwide lacked access to essential health services, and 1 billion faced catastrophic healthcare costs plunging 344 million people into extreme poverty [3]. Populations in eastern Democratic Republic of Congo (DRC) face similar reality. Decades of armed conflict have fostered a persistent crisis [4] that has gradually eroded the health system infrastructure, led to an inadequate distribution of healthcare resources [1], and impeded the government's ability to finance quality and accessible health services for all. In 2021, the DRC government allocated 11.4% of its budget to health, but only 59.6% of this allocation was executed [5]. With less than 15% of the national budget typically allocated to health each year, and low disbursement and execution rates, health facilities commonly rely on patients' direct payment for care to maintain their operations [5, 6]. Out-of-pocket health spending in DRC contributes to over 40% of the overall health budget, 90% of which is through direct payments [5–9]. These financial constraints significantly deter the sick from using healthcare services [10], contributing to the population's growing impoverishment and poor health outcomes.

Moreover, the limited government presence in crisis zones [11] has led to the proliferation of Informal Healthcare Facilities (IHFs). Most IHFs operate marginally within legal frameworks, and the DRC government is often unable to effectively regulate them due to political dynamics that undermine health zone managers' attempts to make IHFs adhere to established regulations [12, 13]. As a result, IHFs often fall short of health sector standards and pose risks to patient safety [12]. Nonetheless, financial constraints and the unrealistic claims made by IHFs draw people to these structures [14]. Strengthening Health Zone (HZ) management is recognized as a pivotal strategy for addressing these challenges, as demonstrated in various contexts, including those similar to eastern DRC [15–18].

In the DRC, the organizational arrangements of the health system are such that leadership must be exercised fully at the lowest level of the health system, through the HZ managers gathered in a Health Zone Managment Team (HZMT) [19]. The HZMT is responsible for managing and coordinating health services, planning interventions, conducting epidemiological surveillance, and supporting primary care structures. They also ensure the effective implementation of national health policies [19, 20]. Previous studies in eastern DRC have documented a significant leadership deficit among HZ's managers in crisis-affected areas. Indeed, while healthcare personnel regularly receive training from various Ministry of Health programs and external partners, a formal government program that aims to systematically and uniformly strengthening the HZ managerial leadership across the entire country is not documented within the scientific literature [12, 21]. In 2015, an evidence-based health system strengthening program named "Renforcement Institutionnel pour des Politiques de Santé basées sur l'Évidence en RD Congo" (RIPSEC: Institutional Strengthening for Evidence-Based Health Policies in DRC) was established in Walungu HZ in the eastern DRC's province of South-Kivu. Unsing the "demonstration health zone" strategy proposed in the DRC's National Health System

Strengthening Strategy though not yet implemented [19], RIPSEC launched transformative activities in Walungu to establish it as a Learning and Research Zone (LRZ). Walungu was selected since it met key selection criteria: it presented a minimally acceptable level of functionality, was geographically accessible, and had favorable security conditions. These criteria ensured that the selected zone could provide a safe, stable, and accessible environment for effective HZMT transformative mentoring. Designated as a LRZ, Walungu was positioned to serve as a model for other HZs, demonstrating viable strategies for developing and improving health services.

This study is based on the hypothesis that leadership development through the "demonstration health zone" approach can effectively address the health challenges in crisis-affected areas. In particular, the study examines the impact of RIPSEC mentorship on Walungu HZMT's leadership capacities in the face of competition from IHFs.

## Methodology

### Study region: Presentation of the case under study

The study was conducted in the Walungu HZ in South-Kivu province, eastern DRC. Walungu is one of 519 operational HZs (also known as Health Districts) that form the foundational layer of the DRC's health system pyramid. Each HZ is structured to provide primary healthcare services via (i) a network of 7 to 25 Health Centers (HCs), each serving its corresponding geographical entity, known as a Health Area (HA), and delivering a minimum package of care, and (ii) a General Referral Hospital (GRH) that offers a complementary package of care. Some HZs also include secondary hospitals or Referral Health Centers that provide a level of care between the first and second lines. The intermediate coordinating level of the health system pyramid comprises 26 Provincial Health Divisions (DPS), one for each of the country's 26 provinces, while the central normative level is headquartered in Kinshasa, the capital city.

Walungu is characterized as a rural HZ with an estimated population of 285,669 in 2019, distributed across 23 HAs (Table 1). The Walungu HZ Central Office 2020 report estimated the health coverage rate at 96%. This is defined as the population living less than 5 km away from an integrated health facility and without significant geographical barriers [20]. Similar to most HZs in eastern DRC, Walungu is a post-conflict zone that continues to experience sporadic attacks on its population by armed rebel groups [1, 22].

### Intervention framework of the RIPSEC program in Walungu

The RIPSEC program, led by a consortium of three Schools of Public Health in the DRC and funded by the European Union, was operational from 2015 to 2019. Its overarching goal was to enhance the use of scientific evidence in health policy development to improve health outcomes. The program was implemented by three DRC's academic public health institutions across three HZs: (i) The University of Lubumbashi in Kisanga HZ in the Haut-Katanga province, (ii) The University of Kinshasa in Gombe-Matadi HZ in the Kongo Central province, and (iii) The Regional Public Health School of the "Université Catholique de Bukavu" (ERSP-UCB) in Walungu HZ in the South-Kivu province [23].

In Walungu, the RIPSEC program included four major "transformation projects" designed to boost the effectiveness of the HZMT, including: (i) leadership strengthening, (ii) formative supervision, (iii) enhancement of referral and counter-referral systems, and (iv) transformation of health facilities into model structures for other HZs.

Leadership development was customized to fit the specific needs of Walungu, focusing on enhancing the HZ's functionality and ensuring efficient coordination of activities. As part of the leadership strengthening component, a mentorship program was led by a public health

**Table 1. General characteristics of the Walungu HZ.**

| | Baseline | RIPSEC Program Years | | | | |
|---|---|---|---|---|---|---|
| | 2014 | 2015 | 2016 | 2017 | 2018 | 2019 |
| **Characteristics** | | | | | | |
| **Population** | 244,684 | 252,269 | 260,362 | 268,434 | 276,755 | 285,669 |
| **Number of Health Areas** | 23 | 23 | 23 | 23 | 23 | 23 |
| **Number of General Referral Hospitals (GRH)** | 1 | 1 | 1 | 1 | 1 | 1 |
| **Number of Referral Health Centers** | 5 | 5 | 5 | 5 | 5 | 5 |
| **Number of Health Centers** | 23 | 23 | 23 | 23 | 23 | 23 |
| **Number of Health Posts** | 3 | 4 | 6 | 6 | 6 | 7 |
| **Number of beds in the GRH** | 200 | 200 | 235 | 233 | 286 | 272 |
| **Health Coverage Rate** | 96.4% | 96.4% | 96.4% | 96.4% | 96.4% | 96.4% |
| **Number of medical doctors** | 9 | 9 | 9 | 9 | 9 | 10 |
| **Number of nurses** | 121 | 127 | 137 | 164 | 153 | 201 |
| **Density of Medical doctors (per 10,000 inhabitants)** | 0.37 | 0.36 | 0.35 | 0.34 | 0.33 | 0.35 |
| **Density of Nurses (per 10,000 inhabitants)** | 4.9 | 5.0 | 5.3 | 6.1 | 5.5 | 7.0 |
| **Beds per 10,000 inhabitants** | 8.2 | 7.9 | 9.0 | 8.7 | 10.3 | 9.5 |

expert and experienced former HZ manager who was working, at the time of his selection, as manager of a health program at provincial level. The Walungu mentor was responsible for organizing weekly HZMT meetings, providing individual coaching to foster greater commitment and participation in management activities. Formative supervision replaced previous supervisory approaches with interactive, on-the-spot error correction and detailed reporting post-visit. The standardization of referral and counter-referral tools aimed to harmonize procedures and improve communication across facilities. Additionally, facilitating learning opportunities between lower and higher-performing facilities aimed to enhance performance indicators and disseminate best practices. All activities were meticulously documented under ERSP-UCB's oversight [24].

## Conceptual framework

The RIPSEC framework (Fig 1) conceptualizes the transformative potential of targeted leadership support within a complex socio-economic, cultural, and administrative context marked by post-conflict recovery. It delineates the process by which RIPSEC mentorship of the HZMT is expected to directly improve the overall HZ functionality and performance. Activities are strategically designed to effectively address the unique challenges encountered in Walungu. Expected outcomes include an increased utilization of curative services and a notable reduction in the negative impacts of IHFs, among other potential improvements.

## Definition of concepts

1. Mentorship: In the context of public health, mentorship refers to a supportive and advisory relationship where an experienced individual (the mentor) guides and assists a less experienced individual (the mentee) in their professional development. The goal is to enhance the mentee's skills, knowledge, and confidence [23]. Within the RIPSEC program, a ERSP-UCB public health expert was assigned to provide ongoing technical support, training, advice, and local coaching to improve HZ managers' leadership and management skills [23, 24].

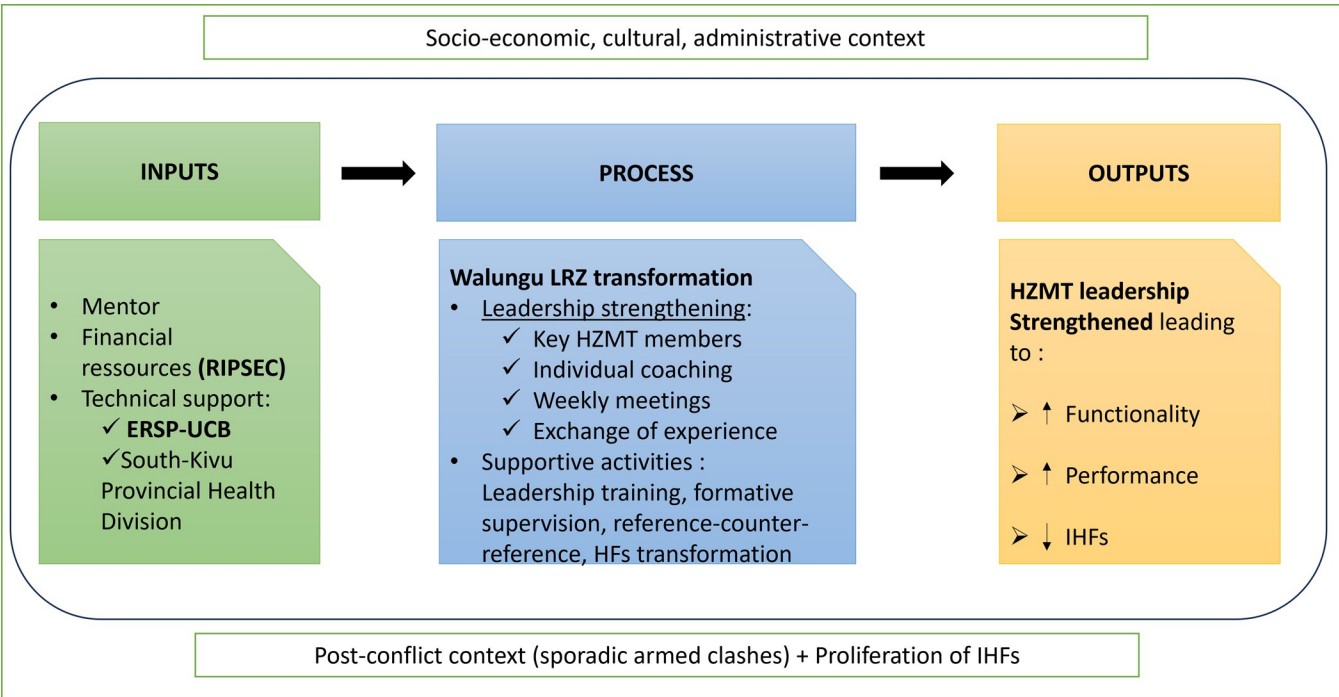

**Fig 1. Conceptual framework of the RIPSEC program intervention in Walungu LRZ.**

2. Learning and Research Zone (LRZ): A LRZ is a designated environment that facilitates learning, research, and the development of improved practices through collaboration between researchers and health managers [13, 25]. In the RIPSEC context, the Walungu LRZ served as a testbed for innovative strategies and interventions aimed at enhancing the healthcare system and generating evidence. The mentorship activities described in this study are integral components of this LRZ's function [23].

3. Health structures: Walungu HZ encompasses a range of health facilities (HF) offering diverse healthcare needs [20]. These include:

- Health Centers: First-line health facilities providing a minimum package of services, including curative, preventive, promotional, and rehabilitation cares.

- Health Posts: First-line health facilities offering a more basic package of care than that provided by Health Centers.

- Referral Health Centers: First-line facilities equipped with maternity units and comprehensive medical consultation services, offering more extensive care than a standard Health Center.

- Walungu General Referral Hospital: Second-line medical facility that offers a complementary package of activities, including the four essential services of Internal Medicine, Surgery, Gynaecology-obstetrics, and Pediatrics.
All these facilities are formal health structures that adhere to the national health protocols. They have been approved by the Ministry of Health and may be either public or private. Among these formal health structures, some are integrated to the national health system. Integrated structures are state-approved facilities that report their health data to the

national health information system. Conversely, private facilities, though state-approved, are not mandated to report their data to the Ministry of Health [19, 26].

4. Informal Healthcare Facilities (IHFs): IHFs are private entities that offer care aimed at curing or alleviating illnesses without the required quality or authorization. They do not adhere to national health protocols and do not report to the national health information system. IHFs fall into two categories:

- Traditional Medicine facilities, which treat illnesses using traditional methods and materials such as plants, leaves, roots, and animal parts not recognized in modern pharmacology.

- Prayer-based facilities, which organize prayer sessions or vigils to seek divine healing.

## Type and period of study

A case study approach was employed to analyze the implementation of the RIPSEC program in the Walungu LRZ. We used a sequential exploratory mixed-methods approach (Table 2), in which the qualitative component followed the quantitative analysis to explore initial findings in greater depth and provide contextual insights. This approach allowed for a comprehensive evaluation of the mentorship program's impact [27]. Retrospective analysis of key health indicators from 2014 (pre-implementation) through the five-year duration of the RIPSEC project (2015–2019), formed the quantitative component. These data were collected continuously from January 1$^{st}$, 2015 to December 23$^{rd}$, 2019 and accessed for analysis in March 2022.

The qualitative component focused on the program's contribution to leadership and governance development. It was structured around an analysis framework derived from the RIPSEC program conceptual framework, and centered on three core themes: (1) the influence of IHFs on the utilization of curative services, (2) the impact of the RIPSEC program on strengthening individual and collective leadership within the HZMT, and (3) the effects of RIPSEC mentorship on the overall HZ performance and functionality. In-depth interviews were conducted with key HZMT members, from November 23rd to December 15th, 2022.

The integration of the quantitative and qualitative components was achieved during the analysis phase. Initial quantitative findings informed the development of the qualitative interview guide, while the qualitative data provided deeper insights and helped interpret the trends observed in the quantitative data. Triangulation of the quantitative and qualitative data enhanced the robustness of our findings, ensuring that insights gained from one method were corroborated by the other [28–30].

## Sampling and data collection

**1) Quantitative component of the study.** To assess the evolution of the HZ's functionality and performance, convenience sampling was employed, focusing on key health indicators

Table 2. Summary of study approaches.

| Approach | Sources | Information sought |
|---|---|---|
| **Quantitative** | Retrospective trend analysis of health indicators from 2014 to 2019. | Evaluation of functionality and performance trends, and the impact of IHFs on service utilization. |
| **Qualitative** | Key informant interviews with members of the Walungu HZMT who served during 2015–2019. | Insights on the impact of mentorship on leadership development and problem management within the zone. |

at the HZ, particularly those related to service utilization in integrated facilities. During the RIPSEC program period, deidentified data were gathered by the program mentor from health reports at health facility and HZ Central Office levels. This data collection process ensured that there was no access to individual identities during and after data collection. The mentor served as a part-time technical assistant and played a pivotal role by coaching the HZMT in organizing the HZ and collecting data monthly. The mentor's physical presence in the LRZ ensured the accuracy of monthly data recording. Observations prior to program launch were restricted to one year (2014) due to the incomplete implementation of the national DHIS2 data system during that period [31] and lack of dedicated personnel to collect data in the HZ. Additionally, archival weaknesses during this period further restricted the reliability of any data prior to 2014. Collected data were stored on a secured electronic folder only accessible to the ERSP-UCB study team.

Performance indicators related to preventive services at first line health facilities and curative services at second-line health facilities were selected to reflect the impact of managerial leadership in health service delivery [4, 15] (Table 3).

To measure HZ functionality, indicators that align with leadership and good governance were chosen, including organizational and managerial capacities that demonstrate effective HZ managers leadership. Such indicators included the frequency of the Board of Directors meeting (HZMT, provincial-level managers, and technical partners from the zone), Management Board meeting (HZ Central Office and GRH managers), HZ Central Office Board of Directors meeting, GRH Board of Directors meeting, HZ Reviews, and HZ Health Development Committee meetings [15].

**Table 3. Performance indicator.**

| Indicator | Definition | Link with Leadership |
|---|---|---|
| **First Line of Care (Health Centre)** | | |
| **First Antenatal Visit Utilization rate** | Numerator: Number of pregnant women who attended their first antenatal visit. Denominator: Total number of pregnant women eligible for that visit. | Reflects the effectiveness of HZ managers in promoting initial prenatal consultations among pregnant women. |
| **Fourth Antenatal Visit Utilization Rate** | Numerator: Number of pregnant women who attended all four antenatal appointments. Denominator: Total number of pregnant women eligible for the fourth antenatal visit. | Demonstrates the ability of managers to ensure continued engagement of pregnant women in ongoing prenatal care. |
| **Third Dose Pentavalent Vaccine Wastage Rate** | Numerator: Difference between the number of children who received the first and third doses of the pentavalent vaccine. Denominator: Number of children who received the first dose of the pentavalent vaccine. | Indicates managerial effectiveness in maintaining adherence to the national child vaccination schedule. |
| **Measles Vaccination Rate** | Numerator: Number of 9-month-old children vaccinated against measles. Denominator: Total number of eligible 9-month-old children. | Shows the success of managers in sustaining long-term vaccination adherence among caregivers throughout the national child vaccine schedule (from 0 to 9 months old) |
| **Second Line of Care (General Referral Hospital)** | | |
| **Curative Services Utilization Rate** | Numerator: Number of new patients treated. Denominator: Number of population-years. | Relates to managers' success in ensuring service availability and accessibility and promoting the utilization of integrated health services. |
| **Rate of Births Attended by Qualified Health Provider** | Numerator: Number of births attended by skilled personnel. | Indicates effectiveness in encouraging professional birth attendance through proactive management and patient education. |
| | Denominator: Total number of births. | |
| **Patients referral rate from health centers to the GRH** | Numerator: Number of patients referred from health centers | Reflects managerial efficiency in adhering to care standards and ensuring appropriate care referrals. |
| | Denominator: Total number of health centers consultations. | |
| **In-hospital Mortality Rate** | Numerator: Number of in-hospital deaths. | Measures the impact of managerial actions on care quality, including protocol implementation, staff training, and outcome monitoring. |
| | Denominator: Total number of hospitalizations. | |

To understand the dynamics of the population that use IHFs, including traditional medicine and prayer groups, we employed a proxy indicator, the number of IHF providers identified in the HAs. We posited that the proliferation of IHFs incentivizes their use [14]; thus, the higher the number of IHFs in a HA, the greater the impact on curative services rates.

We categorized Walungu's HAs into two groups based on the average number of IHFs: "low-IHF HAs" were those HAs with less than the average number of IHFs per HA in Walungu, and "high-IHF HAs" were those at least the average number of IHFs within them. Thus, of the 23 HAs in the zone, 10 were categorized as "low-IHF HAs" and 13 as "high-IHF HAs". This data has been available since 2014 in HZ Central Office's reports and was assumed to be stable through 2019.

**2) Qualitative component.** In-depth interviews were conducted with Walungu HZMT members who had served for at least two consecutive years during or overlapping with the RIPSEC program period. Out of 13 eligible HZMT members identified from the RIPSEC archives, 7 (all male) were successfully contacted and provided informed consent to participate. These 7 participants were considered the most relevant due to their significant roles and experiences with the RIPSEC program. Although we aimed to capture a comprehensive range of perspectives, the consistency and depth of the responses led us to conclude that the data collected was sufficient to address the research questions. The interviews were carried out either face-to-face or remotely, depending on the participants' availability, and verbal informed consent was obtained prior to each session. Each interview lasted about 43 minutes. The interview guide (see S1 File for details) was developed based on the analysis framework, which was informed by the conceptual framework of the study and by the initial quantitative findings. Interviews were conducted by two senior researchers with extensive experience in qualitative methods. All interview recordings and transcripts were securely stored in an electronic folder accessible only to ERSP-UCB study team members.

## Analysis

The data collected in health facilities were organized on MS-Excel spreadsheets (see S1 Database for details), and analyzed using MS-Excel for some, and STATA Version 17 for others. Simple linear regression was used to assess the influence of IHF presence on curative service utilization in 2014 and during the LRZ years (2015–2019), with significance set at $p < 0.05$.

Functionality and performance were evaluated using an internal benchmarking method [32, 33], starting with indicator selection, and scoring (ranging from 0 to 4, where 4 indicates the target achieved). Scores were adjusted based on expected results and predefined standards, allowing for comparative analysis. Walungu's performance in the 13 selected indicators was benchmarked comparatively to standards used by the DRC's health system to assess HZs achievements [4, 32, 34]. Results were illustrated through curves showing changes from 2014 to 2019.

For the qualitative data, a deductive thematic analysis was applied to the interview transcripts, which were anonymized and coded from RP1 to RP7. The thematic framework for the analysis was based on the interview guide and the analysis framework, both of which were developed from the study's conceptual framework and refined by the initial quantitative findings. The analysis focused on three key themes: (i) the mentorship's contribution to leadership, (ii) its effect on the functionality and performance of the HZ, and (iii) its impact on the utilization of curative services in the context of competition with IHFs. Manual coding identified relevant categories, and thematic grouping facilitated theory development and sub-theme identification.

### Ethical considerations

The research protocol for this study was approved by the ethics committee of the Université Catholique de Bukavu under the reference number UCB/CIES/NC/019/2021. The approval was granted under the title "Support for Strengthening the Management and Leadership Skills of the National Malaria Control Program Management Team", which included this sub-study as part of a broader research framework. All stages of the study complied with the relevant guidelines and regulations. For the qualitative interviews, verbal informed consent was obtained directly by the researcher conducting the interviews, who is a member of the research team. The verbal consent process was documented by the researcher immediately following each participant's agreement, noting the date and time of consent. This approach was approved by the ethics committee, taking into account the study's context and the sensitivity of the participant's roles.

## Results

### The use of curative services before and during RIPSEC program

We observed an improvement in the use of curative care services in Walungu's low-IHF HAs from the LRZ years onwards (Fig 2), while there was a deterioration in high-IHF HAs over the same period. Low-IHF HAs consistently showed better rates of curative service utilization. In 2018, there was a sharp decline in the rates for both types of HAs, which subsequently recovered in 2019. It is important to note that the curative services utilization rates remained below 50% across all HAs at all times.

### Influence of IHFs on the use of curative services

Table 4 presents the results from a simple linear regression that assesses the influence of IHFs on the rate of use of curative services in Walungu's HAs. The β value for low-IHF HAs before RIPSEC indicated a trend towards higher use of curative services compared to high-IHF HAs, although this difference was not statistically significant (p = 0.062). During the mentorship years, low-IHF HAs had a statistically significant higher average rate of use of curative services

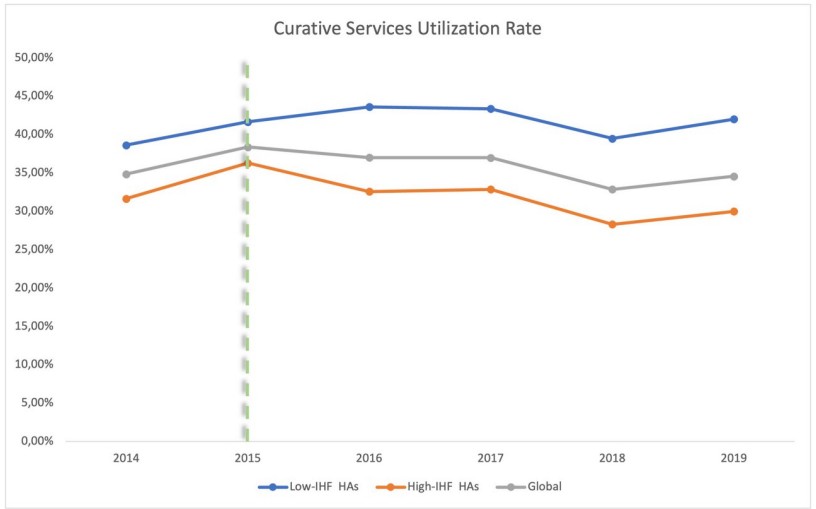

**Fig 2. Curative healthcare services utilization rate trends (the dotted line delimits the periods before and during RIPSEC).**

**Table 4. Use of curative services and IHFs before and during RIPSEC program.**

| Variables | Before RIPSEC (2014) | | During RIPSEC (2015–2019) | |
|---|---|---|---|---|
| | β (95% CI) | p-value | β (95% CI) | p-value |
| **Types of HAs** | | | | |
| Low-IHF HAs | 0.07 (-.001; 0.13) | 0.062 | 0.10 (0.03; 0.17) | 0.007 |
| High-IHF HAs | 0 | | 0 | |

(β = 0.10, p = 0.007) compared to high-IHF HAs. These findings suggest that the presence of IHFs negatively influences the utilization of curative services.

## Walungu's achievements before and during RIPSEC program

A trend analysis in the selected indicators is summarized in Figs 3 and 4. Curative services maintain a constant level throughout the study period, whereas preventive services exhibit positive trends and are consistently recording better performance scores (Fig 3). The management indicators show a general upward trend, albeit with a distinct decline in 2018, followed by a recovery period. The trend analysis of the HZ functionality and performance (Fig 4) indicates an overall improvement during the RIPSEC years compared to 2014. Notably, performance consistently exceeds functionality across the years, with the same sharp decline in 2018 as previously observed.

## HZMT members' experience of RIPSEC mentorship

Our thematic analysis deduced three primary themes, with a fourth unexpected theme emerging: the challenges encountered in sustaining the achievements of the RIPSEC program.

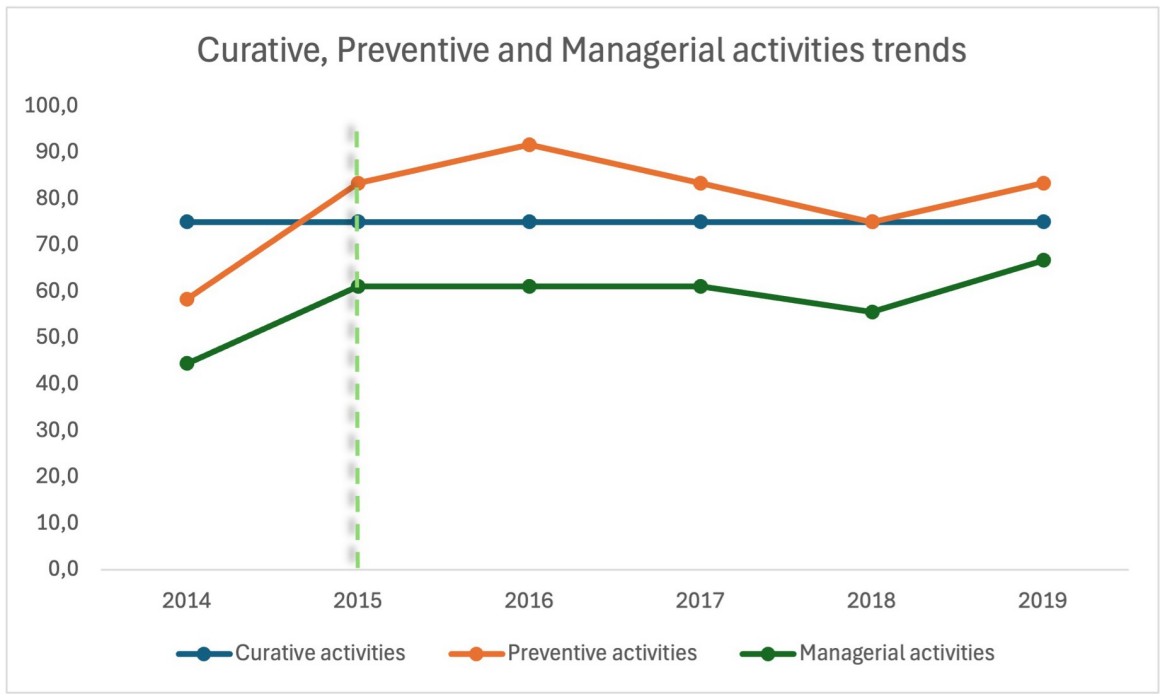

**Fig 3. Preventive, curative and managerial activities trends in Walungu (the dotted line delimits the periods before and during RIPSEC).**

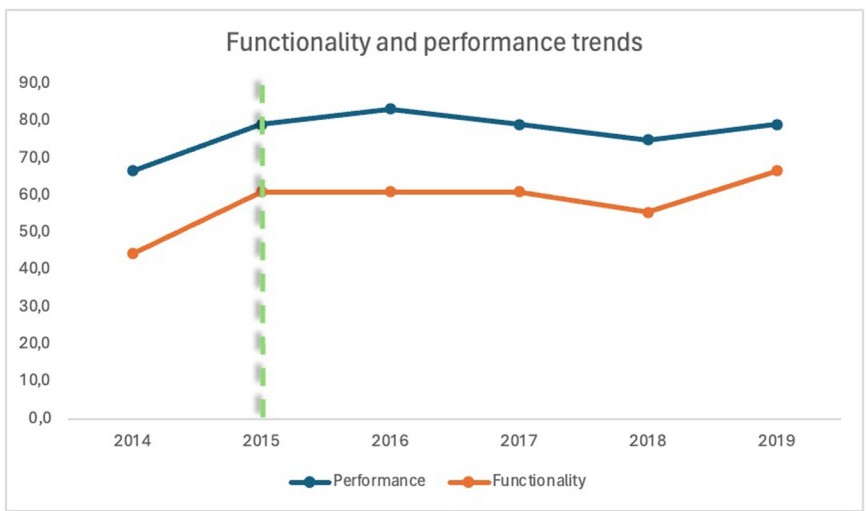

**Fig 4. Functionality and performance trends in Walungu (the dotted line delimits the periods before and during RIPSEC).**

**Leadership under RIPSEC mentorship.**   The interviews highlighted the strengthening of HZMT leadership during the program, emphasizing the significance of a clear vision and effective decision-making. Participants frequently credited the RIPSEC program as a catalyst for this transformation by introducing crucial leadership training and strategies to address identified gaps. Despite financial constraints and limited resources, the training and capacity-building initiatives positively impacted the quality of care and patient management. Respondents noted improvements in accountability, transparency, and coordination across different health management levels.

*"Leadership has really changed the way we run services . . . now there is more accountability and follow-through" (RP1).*

*"Leadership is crucial. Before RIPSEC leadership training, our efforts were scattered." (RP5)*

*"We needed leaders who understood our challenges and could guide effectively." (RP3)*

**Improving health outcomes in the area.**   Respondents recognized staff training and skills development as positive aspects of the RIPSEC program. These initiatives have helped stabilize health indicators in accordance with standards. Specifically, health centers nurse managers were trained to improve the quality of care within their facilities, which improved clinical skills and positively affected health indicators, case management, and team communication through frequent capacity sharing sessions.

*"The training courses have greatly helped our staff to manage cases more effectively" (RP5).*

*"Communication mechanisms have been put in place to enable nurses and referred patients to make contact with the doctors who will see patients in hospital." (RP3)*

*"I can personally say that through the activities that have been carried out with the community, my skills and knowledge have increased as a result of sharing experiences with others.*

*Although I wasn't directly involved, but with the exchanges of experience that we did, it was beneficial for everyone, for the community." (RP7)*

**Use of curative services in competition from IHFs.** Participants discussed how informal care structures, such as prayer houses and traditional healers, significantly attract the population, impacting the use of formal health services, especially in geographically hard to reach HAs. Cultural beliefs and socio-economic challenges as well as poor governance practices have hindered integration of these facilities into a regulated framework. The RIPSEC mentorship attempted to address these issues by developing several initiatives to curb this unregulated competition, leading to improved coordination of care and increased health awareness.

*"When you arrive in almost all the zones at the moment, it's the people from X territory who have unlicensed structures because they currently have a brother from the same territory who is a minister. Now he's encouraged them to open pirate facilities and you, as the zone's chief medical officer, have no power over them. "(RP1)*

*"Within the RIPSEC project, we sought to involve local authorities with the aim of raising awareness of the need to monitor informal care homes." (RP4)*

*"With the presence of these prayer rooms, many health facilities are not able to attract as many patients as they should . . . We have identified and drawn up a list of these problematic facilities in collaboration with community leaders." (RP6)*

*"RIPSEC had trained and informed the community health workers on how to manage and listen carefully to the community's problems . . . I saw the community health workers trained by RIPSEC go out into the community to raise awareness or inform people about their rights but also their advantage of attending the integrated health center." (RP7)*

**Sustaining RIPSEC's achievements.** Challenging disruptive events were reported. Respondents mentioned challenges such as epidemics and the instability of human resources (e.g., the 2018 Ebola Virus epidemic, the death of the GRH medical director, and the departure of the HZ's chief medical officer, in 2018), which significantly slowed down the project's progress. Concerns were raised about resources gaps and the long-term sustainability of improvements.

*"Since RIPSEC, we have seen better use of services, but there are still challenges to be overcome" (RP3)*

*"The lack of funding and equipment is a major challenge for us." (RP2)*

*"In 2018, first there were staff disruptions at hospital level. And then there was the Ebola epidemic, there were the unfortunate events that followed: there were doctors who died in the hospital in the space of two years, and who had benefited from RIPSEC training." (RP5)*

The limited duration of the program (5 years) was frequently cited as a barrier to achieving lasting success and hindered the transmission of LRZ achievements and good practices to other provincial HZs.

*Providers with experience in managing deliveries at the General Referral Hospital were going to work in the two health centers in order to raise the level (of competence of the providers) in*

*the health centers . . . RIPSEC stopped when all 4 projects were not yet well covered. If I had the power, I would relaunch RIPSEC. And as I said, we were a pilot health zone, we were supposed to set an example for the other zones, but unfortunately the project came to an end and the other zones stopped coming . . . (RP1)*

## Discussion

The aim of this study was to explore the impact of a mentorship-based leadership development program on the functionality and performance of a HZ, and on its capacity to resolve specific problems in a crisis context. We focused on the Walungu HZ, faced with the specific problem of the presence of IHFs. We investigated how the transformation of the Walungu into a LRZ under the RIPSEC program's mentorship influenced health outcomes within the HZ. Findings reveal that while the program led to a modest but significant improvement in health service utilization and management capabilities, the persistent influence of IHFs, exacerbated by socio-economic and governance factors, continue to pose significant challenges. Despite the progress made through mentorship, sustainability strategies and continuous support are required to maintain and effectively integrate these initiatives into existing healthcare systems.

### The challenge of IHFs

The rate of formal health service utilization in Walungu remains low between 2014 and 2019 (Fig 2), falling below the standard threshold of 50% [20] and in stark contrast to the provincial average which increased from 34% to 53% during the same period [7].

Trend curves of curative health services utilization rates indicate a decline in formal service usage in areas with a high concentration of IHFs, suggesting that regulatory efforts in these areas may be less effective. However, it is important to note that the difference observed before the intervention was only marginally significant (p = 0.062), indicating that while there was a trend, it was not strongly statistically confirmed. During the mentorship program years, the difference became statistically significant, supporting the hypothesis that IHFs negatively influence the utilization of curative services. Nevertheless, these findings should be interpreted with caution. Further studies, potentially with larger sample sizes or different methodologies, may be required to fully explore the influence of IHFs on utilization of curative services. Respondents indicated that influence peddling often places these informal entities beyond legal oversight, undermining the regulatory authority of HZ managers. The limited decision space available to these managers, constrained by political interference, not only diminishes their leadership prerogatives but also hinders the implementation of strategies aimed at enhancing quality service delivery. Research in similar settings has highlighted the detrimental and pervasive impact of such practices [12, 35]. Additionally, poverty and lack of knowledge are prominent factors driving the community's preference for IHFs. Although the literature suggests that costs can vary widely between public and private sectors, IHFs often offer lower direct costs and more flexible payment options, making them more accessible to financially constrained individuals [6]. Beyond the issue of cost, dissatisfaction with the formal healthcare system, especially for chronic conditions that require substantial psychosocial support and person-centered care, is also a significant factor that drives patients toward IHFs [14, 36–39]. The combination of lower immediate direct costs, flexible payment arrangements, perceived accessibility, and culturally resonant care contributes to the community's preference for IHFs.

Improving healthcare managers' capabilities is a key not only for improving access to care but also for ensuring the quality of care is centered around individual needs, thereby optimizing outcomes despite external pressures.

## LRZ performance and functionality

During the mentorship years, a noticeable improvement was observed in both performance and functionality, compared to the baseline year of 2014. According to respondents, these enhancements were primarily driven by the improvement in leadership and strategic planning fostered through the mentorship. This aligns with findings from several other studies that highlight the positive impact of leadership enhancement on health system performance [15–17, 21]. Our results show fluctuating trends, reflecting the influence of various stability-threatening events both internal and external to the HZ, as detailed by our respondents. Additionally, these fluctuations might also represent other destabilizing factors not accounted for in this study, such as sporadic security crises and disruptions in external funding, both of which heavily influence the system. These factors have been noted by other researchers as significantly affecting health outcomes in HZs [4, 7, 40–42]. It is consistently observed that performance surpasses functionality, likely because healthcare activities tend to receive more support from external partners compared to managerial functions [43, 44].

## Leadership development: The learning site model?

According to our respondents, the RIPSEC mentorship significantly strengthened the leadership skills of HZ managers, enhancing resource management, decision-making, and strategic planning. This progress aligns with mechanisms identified in other sub-Saharan contexts, where increased self-efficacy and perceived autonomy among managers lead to positive outcomes within supportive and conducive working environments [45, 46]. These findings are consistent with the realist evaluation approach, which posits that the effectiveness of support depends on specific contextual factors, actors, and mechanisms, encapsulated in the 'Intervention-Context-Actor-Mechanism-Outcome' (ICAMO) configuration [47, 48]. By fostering psychological safety and trust, RIPSEC mentorship has effectively facilitated knowledge transfer, merging theoretical insights with practical field challenges, and tailored strategies to meet Walungu's unique needs. The importance of real-time, on-site learning for governance, as highlighted in studies from Kenya and South Africa, complements our observations of enhanced decision-making and strategic planning in Walungu [49]. Similarly, the learning site model, which supported micro-practices of governance, has provided a dynamic setting in which health managers engage in informed decision-making. This method enhances understanding and supports emergent system changes, demonstrating the value of site-based, engaged research programs, as further evidenced by a systematic review of district health system management [50].

## Sustaining achievements

Our study revealed challenges in maintaining the improvements achieved after the RIPSEC program ended. Immediately after the implementation of the RIPSEC mentorship, initial improvements in service use were observed. However, in 2018, the trend curves indicated a significant disruption in progress due to events such as the loss of skilled human resources and an epidemic emergency, as reported by our respondents. Similar constraints have been noted in other low-income contexts, where the success of interventions like RIPSEC not only relies on initial outcomes but also on the health system's resilience to acute emergencies and the sustainability of funding and managerial practices [4, 23, 50, 51]. These findings suggest that while programs like RIPSEC can positively impact health outcomes, they are insufficient on their own to overcome all challenges. The lack of sustainable funding strategies, reliance on external aid, and instability of the health workforce could undermine the progress made in LRZ-type strengthening strategies. Experiences from other Sub-Saharan African LRZs indicate

that long-term success and scalability depend on continuous engagement, adaptation of best practices, and integration of research into everyday managerial activities [23] The qualitative assessment of LRZs underscores the need to promote reflexivity in managerial decisions and to focus research on immediate LRZ challenges to foster transformative mentorship. Future studies should explore approaches that integrate these initiatives into existing healthcare systems and ensure ongoing support.

## Limits of the study

Although this study has provided valuable insights into the impact of the RIPSEC program on leadership strengthening and health service improvement in the Walungu HZ, it is not without limitations. Firstly, the retrospective design of the quantitative analysis constrains our capacity to definitively ascertain causal relationships between the RIPSEC interventions and the observed enhancements in health indicators. Secondly, the qualitative component, while providing depth through participant perspectives, may be subject to retrospective or social desirability bias, potentially affecting the objectivity of the responses [52]. Thirdly, focusing exclusively on a single HZ limits the extent to which the findings can be generalized to other settings. Despite these considerations, the study significantly advances our understanding of the impact of health leadership development programs in contexts of crisis.

## Conclusion

This study demonstrates that the RIPSEC program positively influenced leadership enhancement and health service improvement in the Walungu HZ. Despite these gains, persistent financial challenges and the prevalent IHFs remain obstacles that the program alone cannot overcome. Mentorship under RIPSEC improved resource management, decision-making, and strategic planning, contributing to a modest increase in the formal health services utilization. Yet, the reliance on external support and workforce instability have compromised the long-term sustainability of these improvements. This underscores the necessity for durable strategies to embed such initiatives within the existing health system framework. While the study reaffirms the critical role of leadership in public health, particularly in crisis-affected settings, it also highlights that mentorship alone is insufficient to fully address the managerial challenges HZs face. Support from external aid remains essential in crisis areas to progress towards universal health coverage.

## Supporting information

**S1 File. IDI guide.**
(PDF)

**S1 Database. Quantitative database.**
(ZIP)

## Acknowledgments

We extend our thanks to all those who contributed to the successful completion of this study. We are particularly grateful to the Walungu's HZ staff and management team for their cooperation and assistance throughout the research process. Special thanks to the RIPSEC program for their invaluable mentorship which were pivotal in conducting this research. Our gratitude also goes to staff of researchers of the ERSP-UCB for their support for this study.

## Author Contributions

**Conceptualization:** Rosine Bigirinama, Ghislain Bisimwa, Pacifique Mwene-Batu, Abdon Mukalay, Denis Porignon, Albert Tambwe.

**Data curation:** Rosine Bigirinama, Aimé Cikomola, Jean-Corneille Lembebu, Christian Chiribagula.

**Formal analysis:** Rosine Bigirinama, Jean-Corneille Lembebu, Christian Chiribagula.

**Investigation:** Rosine Bigirinama, Aimé Cikomola, Jean-Corneille Lembebu.

**Methodology:** Rosine Bigirinama, Ghislain Bisimwa, Samuel Makali, Janvier Barhobagayana, Albert Tambwe.

**Project administration:** Ghislain Bisimwa, Albert Tambwe.

**Resources:** Aimé Cikomola, Jean-Corneille Lembebu, Christian Chiribagula.

**Software:** Jean-Corneille Lembebu, Christian Chiribagula.

**Supervision:** Ghislain Bisimwa, Albert Tambwe.

**Validation:** Ghislain Bisimwa, Janvier Barhobagayana, Abdon Mukalay, Albert Tambwe.

**Visualization:** Samuel Makali, Christian Chiribagula.

**Writing – original draft:** Rosine Bigirinama.

**Writing – review & editing:** Rosine Bigirinama, Ghislain Bisimwa, Samuel Makali, Janvier Barhobagayana, Pacifique Mwene-Batu, Abdon Mukalay, Denis Porignon, Albert Tambwe.

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
