## [Decision Letter · Decision Letter 0]

16 Jul 2024

PGPH-D-24-01193

The effect of mentorship as a means of strengthening leadership in the health system at the operational level: a case study of the Walungu rural health zone in the eastern Democratic Republic of Congo

Dear Dr. Bigirinama,

Thank you for submitting your manuscript to PLOS Global Public Health. After careful consideration, we feel that it has merit but does not fully meet PLOS Global Public Health’s publication criteria as it currently stands. Therefore, we invite you to submit a revised version of the manuscript that addresses the points raised during the review process.

Dear authors,

We invite you to consider all comments by the reviewers and offer a point-by-point response to each comment. Please feel free to respond to the comments made in French (in a separate Word document provided by the third reviewer), in the language of your preference (French or English).

Please also make sure to follow standard guidelines for your bibliographical references and the format of your paper.

Best wishes and looking forward to the next version,

Lara Gautier

We look forward to receiving your revised manuscript.

Kind regards,

Lara Gautier

Academic Editor

Journal Requirements:

1. In the ethics statement in the Methods, you have specified that verbal consent was obtained. Please provide additional details regarding how this consent was documented and witnessed, and state whether this was approved by the IRB

2. In the online submission form, you indicated that The datasets analyzed during the current study are available from the corresponding author on reasonable request.. 

3. Uploaded as supplementary information.

3. Please provide a/amend your detailed Financial Disclosure statement. This is published with the article. It must therefore be completed in full sentences and contain the exact wording you wish to be published.

**Please only choose the relevant sentences from below**

a. Please clarify all sources of funding (financial or material support) for your study. List the grants (with grant number) or organizations (with url) that supported your study, including funding received from your institution. 

b. State the initials, alongside each funding source, of each author to receive each grant.

c. State what role the funders took in the study. If the funders had no role in your study, please state: “The funders had no role in study design, data collection and analysis, decision to publish, or preparation of the manuscript.”

d. If any authors received a salary from any of your funders, please state which authors and which funders.

Reviewers' comments:

Reviewer's Responses to Questions

**Comments to the Author**

1. Does this manuscript meet PLOS Global Public Health’s publication criteria? Is the manuscript technically sound, and do the data support the conclusions? The manuscript must describe methodologically and ethically rigorous research with conclusions that are appropriately drawn based on the data presented.

Reviewer #1: Partly

Reviewer #2: Yes

Reviewer #3: Partly

2. Has the statistical analysis been performed appropriately and rigorously?

Reviewer #1: I don't know

Reviewer #2: Yes

Reviewer #3: Yes

3. Have the authors made all data underlying the findings in their manuscript fully available (please refer to the Data Availability Statement at the start of the manuscript PDF file)?

Reviewer #1: No

Reviewer #2: Yes

Reviewer #3: Yes

4. Is the manuscript presented in an intelligible fashion and written in standard English?

Reviewer #1: Yes

Reviewer #2: Yes

Reviewer #3: Yes

5. Review Comments to the Author

Reviewer #1: I congratulate the authors of the article on their high-quality work and their sense of integrity in the conclusion. However, I am still thirsty for data on the indicators they claim to have used to measure the impact of RIPSEC's support. Of the 8 performance indicators announced, only the results for the curative service utilization rate were presented and discussed with significance (Table 4). I would suggest also presenting the results of the other indicators individually, and then summarize them as you have done. In addition, you should describe the method used to aggregate these other indicators. Done as proposed, the article can be published.

Reviewer #2: This is a well written manuscript.

The authors have made a clear description of the setting.

The results are presented in a understanding way.

The discussion showed a good integration of the results of the two component of the mixed methods research.

For your attention:

Could the authors indicate how and when they proceeded to the integration of data from the two components of the study?

Could the authors indicated how they decided to stop the selection of the participants for the qualitative phase. Was the saturation of data attained?

These are two sections that could be erased to ease the readability: definition of concepts (lines 175-211) and Note (lines 286-300).

I would recommend that the interview guide be attached as a supplementary file.

Reviewer #3: Comment peut-on parler d’une influence négative des établissements de soins de santé informels sur l’utilisation des soins dans une zone de santé qui n’a qu’1 centre de santé dans une aire de santé pendant que le secteur privé occupe tout espace en termes de couverture sanitaire (accessibilité) ?

6. PLOS authors have the option to publish the peer review history of their article (what does this mean?). If published, this will include your full peer review and any attached files.

**Do you want your identity to be public for this peer review?** For information about this choice, including consent withdrawal, please see our Privacy Policy.

Reviewer #1: **Yes: **Dieudonné TSHISHI BAVUALA

Reviewer #2: No

Reviewer #3: **Yes: **KIETO ZOLA, Eddy

---

## [Decision Letter · Decision Letter 1]

23 Oct 2024

PGPH-D-24-01193R1

The effect of mentorship as a means of strengthening leadership in the health system at the operational level: a case study of the Walungu rural health zone in the eastern Democratic Republic of Congo

Dear Dr. Bigirinama,

Thank you for submitting your manuscript to PLOS Global Public Health. After careful consideration, we feel that it has merit but does not fully meet PLOS Global Public Health’s publication criteria as it currently stands. Therefore, we invite you to submit a revised version of the manuscript that addresses the points raised during the review process.

Dear authors,

Thank you for this revised version. I believe you have adequately addressed the reviewers' comments, except the comment on the mixed methods research design.

While you specified that your study employs a sequential mixed methods research design, you also need to explain what type of sequence you chose and why (i.e., sequential explanatory or exploratory?). I strongly recommend that you read mixed methods research handbooks/chapters on the matter, and that you cite these works as references. Please read and cite: Pluye, P., Bengoechea, E. G., Granikov, V., Kaur, N., & Tang, D. L. (2018). Tout un monde de possibilités en méthodes mixtes: revue des combinaisons des stratégies utilisées pour intégrer les phases, résultats et données qualitatifs et quantitatifs en méthodes mixtes. Oser les défis des méthodes mixtes en sciences sociales et sciences de la santé, 28. URL: https://d1wqtxts1xzle7.cloudfront.net/56141828/Cahier_scientifique_117__Methodes_mixtes_VF_20180322-libre.pdf?1521817833=&response-content-disposition=inline%3B+filename%3DOser_les_defis_des_methodes_mixtes_en_sc.pdf&Expires=1729611524&Signature=HUKq-yPOY4-aqkgF~WUJqeDNkNvjsdUAM5KjWIgFv-wrbR30xkKSDnkA1BblrPUQ3Ht9omzydUwfe29xI8J31oyNsq5m7iva4~Gjo9VVB~jc5455Vpj0QOo8BivEgzeAxGn8kv8yiz3CTb~xiUSpd6uIMxiiv31KyMldUTa00eFd6VO5r3NLKOBICQCX3F3CC6ZoJKVi73x1NjQ24d-TOvJLvPyrvfwgnmhYMwagV9fPcOA3VraDX0DZUb0HEWgy-FfG1JsehS5CxUF0kHJHg47wIOhwO6am3QinSJT0PbIYtBOFlRo4mZGC9hwU24Tb7YTfXeTwH0T9VjO4gr6hWQ__&Key-Pair-Id=APKAJLOHF5GGSLRBV4ZA#page=28

and/or the following chapter: https://scienceetbiencommun.pressbooks.pub/evalsantemondiale/chapter/integration/

In addition, many of the contents are wordy due to French-English translations. I recommend that you ask the service of a professional English language editor to finalise your work.

Thank you and good luck,

Lara Gautier, Academic Editor

We look forward to receiving your revised manuscript.

Kind regards,

Lara Gautier

Academic Editor

Journal Requirements:

Reviewers' comments:

Reviewer's Responses to Questions

**Comments to the Author**

1. If the authors have adequately addressed your comments raised in a previous round of review and you feel that this manuscript is now acceptable for publication, you may indicate that here to bypass the “Comments to the Author” section, enter your conflict of interest statement in the “Confidential to Editor” section, and submit your "Accept" recommendation.

Reviewer #1: All comments have been addressed

Reviewer #2: All comments have been addressed

2. Does this manuscript meet PLOS Global Public Health’s publication criteria? Is the manuscript technically sound, and do the data support the conclusions? The manuscript must describe methodologically and ethically rigorous research with conclusions that are appropriately drawn based on the data presented.

Reviewer #1: Yes

Reviewer #2: Yes

3. Has the statistical analysis been performed appropriately and rigorously?

Reviewer #1: I don't know

Reviewer #2: Yes

4. Have the authors made all data underlying the findings in their manuscript fully available (please refer to the Data Availability Statement at the start of the manuscript PDF file)?

Reviewer #1: Yes

Reviewer #2: Yes

5. Is the manuscript presented in an intelligible fashion and written in standard English?

Reviewer #1: Yes

Reviewer #2: Yes

6. Review Comments to the Author

Reviewer #1: (No Response)

Reviewer #2: No additional comments.

7. PLOS authors have the option to publish the peer review history of their article (what does this mean?). If published, this will include your full peer review and any attached files.

**Do you want your identity to be public for this peer review?** For information about this choice, including consent withdrawal, please see our Privacy Policy.

Reviewer #1: No

Reviewer #2: **Yes: **Jean-Pierre FINA LUBAKI

---

## [Editor Report · Decision Letter 2]

20 Nov 2024

Mentorship to strengthen health system leadership: a case study of the Walungu rural health zone in the eastern Democratic Republic of Congo

PGPH-D-24-01193R2

Dear Mrs Bigirinama,

We are pleased to inform you that your manuscript 'Mentorship to strengthen health system leadership: a case study of the Walungu rural health zone in the eastern Democratic Republic of Congo' has been provisionally accepted for publication in PLOS Global Public Health.

Best regards,

Lara Gautier

Academic Editor
